# AESTHETICNET: REDUCING BIAS IN FACIAL DATA SETS UNDER ETHICAL CONSIDERATIONS

## ABSTRACT

Facial Beauty Prediction (FBP) aims to develop a machine that can automatically evaluate facial attractiveness. Usually, these results were highly correlated with human ratings, and therefore also reflected human bias in annotations. Everyone will have biases that are usually subconscious and not easy to notice. Unconscious bias deserves more attention than explicit discrimination. It affects moral judgement and can evade moral responsibility, and we cannot eliminate it completely. A new challenge for scientists is to provide training data and AI algorithms that can withstand distorted information. Our experiments prove that human aesthetic judgements are usually biased. In this work, we introduce AestheticNet, the most advanced attractiveness prediction network, with a Pearson correlation coefficient of 0.9601, which is significantly better than the competition. This network is then used to enrich the training data with synthetic images in order to overwrite the ground truth values with fair assessments.

We propose a new method to generate an unbiased CNN to improve the fairness of machine learning. Prediction and recommender systems based on Artificial Intelligence (AI) technology are widely used in various sectors of industry, such as intelligent recruitment, security, etc. Therefore, their fairness is very important. Our research provides a practical example of how to build a fair and trustable AI.

## 1 MOTIVATION

In 2016 *Beauty.AI*, a Hong-Kong based technology company, hosted the first international beauty contest judged by artificial intelligence (beauty.ai, 2016) but the results were heavily biased, for example, against dark skin (Levin, 2016) subjects. "Machine learning models are prone to biased decisions, due to biases in data-sets" (Sharma et al., 2020). Biased training data potentially leads to discriminatory models, as the datasets are created by humans or derived from human activities in the past, for example hiring algorithms (Bogen, 2019). The reason for racist and discriminatory tendencies must be identified. As the learning algorithms become more complex, understanding why the decisions are made, or even how, prove to be nearly impossible (Bostrom & Yudkowsky, 2018). Therefore, the development of non-biased and fair training data and AI algorithms (defined by the European Commission High-Level Expert Group on Artificial Intelligence (European Commission High-Level Expert Group on Artificial Intelligence [AI HLEG], 2019)) is a new and increasingly complex challenge for scientists around the world (Bellamy et al., 2018). The specific field of aesthetic judgement is especially vulnerable to being biased, as aesthetic judgement itself is already a subjective rating (Richmond, 2017).

The purpose of facial beauty prediction (FBP) research is to classify images mimicking subjective human judgements. Investigations related to machine perception in a ground-truth free setting show that the data source depends on the measurement of human perception (Prijatelj et al., 2020). Therefore, artificial networks need a process to determine labels of the average person's judgement. Our data analysis has already proven that people consider their own ethnicity to be more attractive than others (Gerlach et al., 2020), this is the major bias in our experiments and within our dataset. With this tendency, it becomes difficult to generate input data to train a machine-learning algorithm, which assesses a person's attractiveness without bias.

This work not only helps to achieve moral enhancement through AI (see appendix B.1), but also helps eliminating social problems with this new technology (see appendix B.2).

## 2 STATE OF THE ART

While research on the estimation of images or portraits is not a new trend, it has gained increasing attention since the emergence of artificial intelligence (Zhang & Kreiman, 2021). Although, for many applications like autonomous driving, or image classification, AI undoubtedly is the best solution, applications that are affected by unconscious bias, like beauty prediction (Dornaika et al., 2020), tend to reflect bias that is likely to be prevalent within given datasets. Especially, when people subjective preferences play a role, such as in attractiveness judgement (Shank & DeSanti, 2018) or human resource evaluation (Lloyd, 2018), bias is almost certain to happen. Carrera (2020) conducted a piece of research on the implication of racism in image databases, that analysed the association of aggressiveness, kindness, beauty and ugliness with different images and found that the decisions of many people are affected by subconscious racism. Since researchers are aware of such effect, they found different ways to reduce subconscious bias in machine learning. Since the problem originates from the given databases, either the databases, or the training need to be changed.

The possibilities to change the databases include adding data, also referred to as *fair pre-processing* (Bellamy et al., 2018), either by selection or augmentation to insert underrepresented samples. While, deleting images is usually a bad idea, since it increases the chances of the network overfitting, it could theoretically be used to eliminate overrepresented images. On the other side, training can be altered by selecting only images, that do not increase the variance of each class currently used as training input. For example, variational autoencoders can be used to extract the features of the image, to later determine their variance, and only select images as input that do not increase the variance within given classes. Bellamy et al. (2018) also describe a third method, they called *fair post-processing*. Since their pipeline aimed to create debiased databases, the post-processing step is usually not applicable for most machine learning applications not creating databases.

## 3 BIASED AI

### 3.1 BIAS FROM HUMAN INDICATIONS

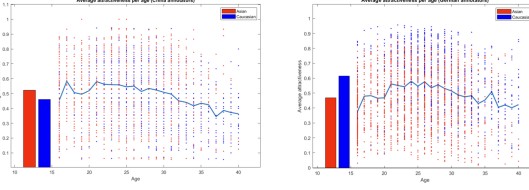 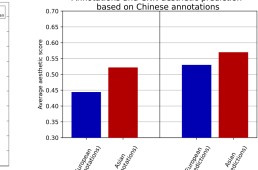 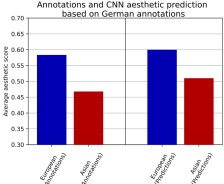

Figure 1: Chinese and European annotations, the red bar represents the score of Asian, the blue bar represents the score of European faces.

Figure 2: AestheticNet is trained on German or Chinese annotations only. The trained network follows the bias from the annotations.

First, we propose hypothesis 1: The results of the evaluation of the attractiveness of female pictures in the Asia-Europe data set by annotators in China and Germany are implicitly biased. We use our latest data set, which includes a total of 12,034 images of people from different social and ethnical backgrounds, with a total of 5.4 million annotations. Chinese and German participants have rated the pictures in the data set. We then have a comparison result to prove whether hypothesis 1 is true and mark this evaluation result as a ground truth. Figure 1 and fig. 2 confirm the statement of the first hypothesis. Further more, in this process, the results of our research also shows that aesthetic bias is not only related to ethnic background, but also related to age which also has been proven by other researchers (Gerlach et al., 2020), (Akbari et al., 2020).

### 3.2 AI TAKES ON HUMAN BIAS

We propose hypothesis 2: artificial intelligence will copy the human bias. We use convolutional neural networks (CNN) to predict facial aesthetic scores and introduce AestheticNet.

**Related Work.** With the introduction of CNNs and large-scale image repositories, facial image and video tasks get more powerful (Krizhevsky et al., 2017; Zeiler & Fergus, 2013; Deng et al.,

2009). Xie et al. (Xie et al., 2015a) present the SCUT-FBP500 dataset, containing 500 Asian female subjects with attractiveness ratings. Since "FBP is a multi-paradigm computation problem" the successor SCUT-FBP5500 (Liang et al., 2018) is introduced in 2018, including an increased database of 5500 frontal faces with multiple attributes: male/female, Asian/Caucasian, age, beauty score. Liang et al. (2018) have evaluated their database "using different combinations of feature and predictor, and various deep learning methods" on AlexNet (Krizhevsky et al., 2017), ResNet-18 (He et al., 2015) and ResNeXt-50 and achieved the Pearson Correlation *PC:* 0.8777; mean average error *MAE:* 0.2518; root-mean-square error *RMSE:* 0.3325 as a benchmark. In summary it can be said that all deep CNN models are superior to the shallow predictor with hand-crafted geometric feature or appearance feature (Liang et al., 2018).

**Benchmark Dataset.** The SCUT-FBP 5500 data set is a small data set for deep learning tasks. Therefore, it is an even greater challenge to train soft features like aesthetic or beauty. In order to measure the accuracy of the network and to be comparable to recent experiments in facial beauty prediction, we calculate the Pearson correlation coefficient (PC), mean absolute error (MAE) and root mean square error (RMSE).

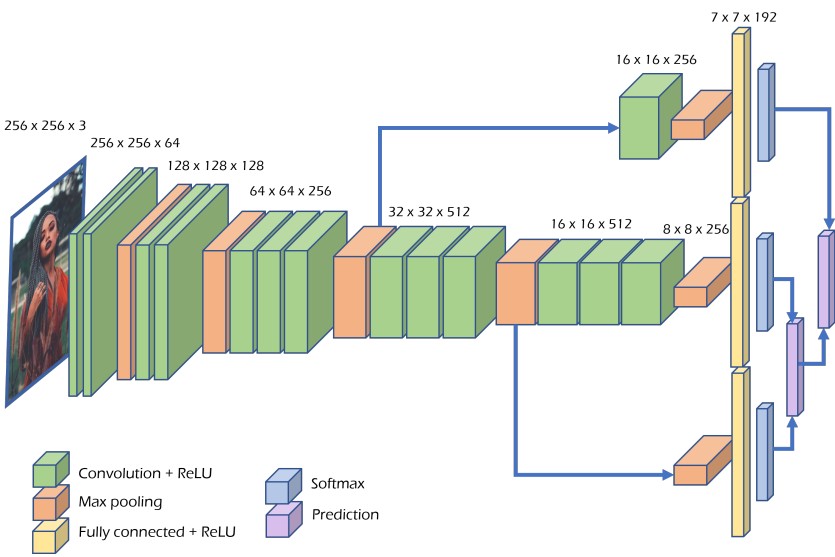

Figure 3: The architecture of AestheticNet is based on the VGG Face architecture and is expanded by two separate skip connections. At the end, the predictions of the differently convoluted feature vectors are added together.

**AestheticNet predictor architecture.** The VGG Face architecture (Simonyan & Zisserman, 2015) is the basis of our AestheticNet. Inspired by an idea of the paper from Shelhamer et al. (2017) we then add modifications to the network by exploiting feature maps from the third and fourth convolution block. Since the size of the features maps differ from the size of the resulting feature vector, we implement an additional max pooling layer to achieve the wanted output. For the predictions of the network, we concatenate the softmax results into a single feature vector as shown in fig. 3.

Our proposed network achieves a Pearson correlation coefficient of 0.9601, which indicates an almost linear correspondence between annotations and predictions. Our training results have a very high accuracy and outperform state-of-the-art results. The normalised mean square error is 3.896% and the normalised root mean square error is 5.580%. These are measurements of the average error of the predicted labels, which are used to evaluate the accuracy of the network. The results are normalised because there are different datasets with different score ranges.

**Reannotation of SCUT-FBP5500 dataset.** Since 2013, for our study of facial aesthetics, we conducted online surveys on multiple image datasets (mentioned in table 2) where thousands of students and their relatives participated. With this process we have been able to gather enough data to train a convolutional neural network with the goal to improve facial beauty prediction. During

Table 1: Comparison of prediction accuracy on SCUT-FBP5500

| Architecture | PC | nMAE [%] | nRMSE [%] |
|---|---|---|---|
| AlexNet [1] | 0.8298 | 7.345 | 9.548 |
| AlexNet [2] | 0.8634 | n/a | n/a |
| ResNet-18 [3] | 0.8513 | 7.045 | 9.258 |
| ResNeXt-50 [4] | 0.8777 | 6.295 | 8.313 |
| HMTNet [5] | 0.8783 | 6.2525 | 8.158 |
| AaNet [6] | 0.9055 | 5.590 | 7.385 |
| P-AaNet [7] | 0.8965 | 5.713 | 7.588 |
| 2M BeautyNet [8] | 0.8996 | n/a | n/a |
| EfficientNetB3 based AestheticNet (ours) | 0.9011 | 5.841 | 7.663 |
| VGG-Face based AestheticNet (ours) | 0.9363 | 4.400 | 6.261 |
| **AestheticNet (ours)** | **0.9601** | **3.896** | **5.580** |

training convolutional neural networks (CNN) on this data, we recognised a large bias in this data. This led us to evaluate the annotations from Chinese and German universities and take a closer look at the bias. Our null hypothesis was that there is no bias in dependency of the ethical group, the proof for the presence of bias was done by reductio ad absurdum.

In null hypothesis significance testing, the p-value is the probability of obtaining test results at least as extreme as the results actually observed, under the assumption that the null hypothesis is correct (Aschwanden, 2015). The precise calculation of the p-value in this experiment is difficult because the factorials raise too high, to be reasonably computed on the thousands of labelled values. We calculate the p-value on 300 representative annotations which lead to a p-value of approximately 0.063%, therefore it is safe to say the null hypothesis can be rejected and we do have an ethical bias.

## 4 TRAINING OF UNBIASED AI

In general, there are three main paths to reach the goal of unbiased predictions: fair pre-processing, fair in-processing and fair post-processing (Bellamy et al., 2018). Within this paper, we present two approaches based on those paths to train an unbiased network with biased data, for FBP. The first approach relies on data pre-processing before training to introduce fairness, we call it "balanced training". The second approach relies on a categorical cross entropy loss function, for the network to learn the bias and decrease it. Those processes are explained in the following sections.

### 4.1 DATASET AND GAN IMAGES

Machine learning has evolved in the past decades and stands out due to the fact that the knowledge in the system is not provided by experts. Facial beauty prediction (FBP) that is consistent with human perception, is a significant visual recognition problem and a much-studied subject in recent decades. Eisenthal et al. (2006) and (Kagian et al., 2008) were among the first to publish their research about automatic facial attractiveness predictors and supervised learning techniques, based on the extraction of feature landmarks on faces. We analysed the data that we gathered with our Analysis Toolbox and could measure a significant bias within the prediction of aesthetics through different ethnicities. Therefore, training a network with the goal to create unbiased results is still a challenge in deep learning tasks. In the following we will first describe our data set blend and the accompanying Analysis Toolbox and we explain how we used a GAN to create artificial portraits with European and Asian ethnicities.

Starting in 2017, we used the Asian-European-dataset SCUT-FBP (Xie et al., 2015b; Liang et al., 2018) to evaluate biased annotations from Chinese and German universities. The results proved the assumption that German students favour images of European women and vice versa Chinese students rate Asian portraits higher. Since the SCUT-FBP 5500 dataset is a small dataset for deep learning tasks, we use data augmentation methods to enlarge the sample size of the training set by generating GAN images with either Asian or European or mixed images as input and new synthesised images as output. This augmentation method proves superior to geometric transformations like cropping

and rotating. All images are preprocessed, by normalisation methods to harmonise face pose, facial landmark positions and image size.

Table 2: Our dataset blend and annotations

| datasets | | | annotations | | | | | | | | | | |
|---|---|---|---|---|---|---|---|---|---|---|---|---|---|
| since | name | faces | age | gender | ethnic | height | weight | sports | glasses | attractiveness | complexion | hair colour | hair style |
| 2013 | MCSO Criminals [9] | 750 | ✓ | ✓ | ✓ | ✓ | ✓ | ✗ | ✗ | ✓ | ✗ | ✓ | ✗ |
| 2013 | Olympics [10] | 1914 | ✓ | ✓ | ✓ | ✓ | ✓ | ✓ | ✗ | ✓ | ✗ | ✗ | ✗ |
| 2016 | LFW [11] | 1578 | ✓ | ✗ | ✓ | ✗ | ✗ | ✗ | ✓ | ✓ | ✗ | ✗ | ✗ |
| 2018 | SCUT-FBP* [12] | 2750 | ✓ | ✓ | ✓ | ✗ | ✗ | ✗ | ✗ | ✓ | ✗ | ✗ | ✗ |
| 2020 | synthesised Eurasians (ours) | 2942 | ✓ | ✓ | ✓ | ✗ | ✗ | ✗ | ✓ | ✓ | ✓ | ✓ | ✓ |
| 2021 | FairFace [13] | 2100 | ✓ | ✓ | ✓ | ✗ | ✗ | ✗ | ✗ | ✓ | ✗ | ✗ | ✗ |
| | Σ 12034 | | ✓ | | ✓ | | | | | ✓ | | | |

For the purpose of a thorough analysis, we blend multiple datasets in the domain of facial aesthetics together. Our complete set of databases which is described in table 2, consists of multiracial and multiethnic individuals. In total, this data set includes 12,034 portrait images from persons of different ethnicities with individual social backgrounds. These images are labelled and annotated in surveys over a period of 8 years with a total number of 5.4 million annotations. Additionally, recently we add the FairFace (Kärkkäinen & Joo, 2019) database, which includes male and female portraits of seven different ethnic groups.

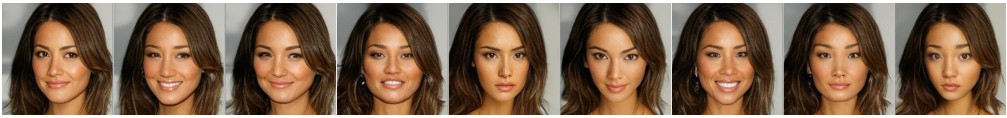

Figure 4: StarGAN v2 generated Eurasians. From left to right: 90%, 80%, 70%, 60% European, half/half, 60%, 70%, 80% 90% Asian

The synthesised Eurasians images are artificially generated with StarGAN v2 (Choi et al., 2020) to determine the influence of the biased view of annotators on aesthetics of persons from different ethnicities. We used different customised input for the source and reference images to control the amount of ethnic admixture. Figure 4 shows one exemplary set of images for the Eurasians dataset.

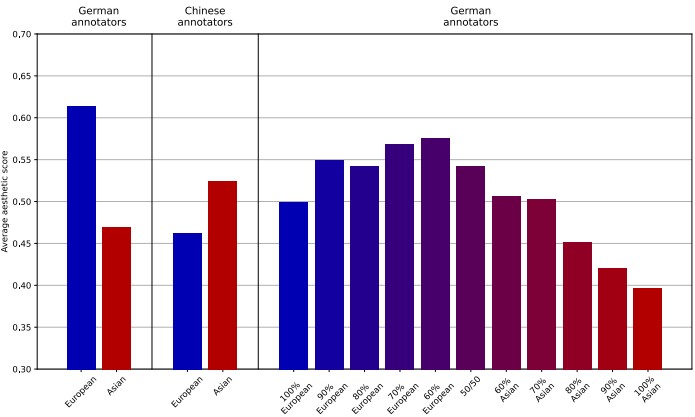

Figure 5: Unconscious bias towards ethnic aesthetic of either German or Chinese annotators. Left: average aesthetic score on SCUT-FBP by German annotators, middle: average aesthetic score labelled by Chinese students, right: aesthetic scores on the Eurasian dataset annotated by German students.

After annotating the dataset, the unconscious bias in the annotations can be uncovered. Figure 5 shows the biased average score of our networks on the SCUT-FBP dataset and the Eurasian dataset.

Figure 6 illustrates the analysis on the distribution of aesthetic score and age for Asians, Europeans and three mixed-racial subgroups. The different group annotation points are displayed in different colours. We calculate the following metrics for each group cluster $i$: Horizontal dashed lines are average attractiveness values $\overline{a}_i$. Vertical dashed lines are average age values $\overline{y}_i$. As can be seen, the interval of $\overline{a}_i$ has a small span, yet however the interval of $\overline{y}_i$ has a significantly larger span. Each $\overline{a}_i$ and $\overline{y}_i$ values intersection point forms an per group attractiveness-age-factor $AAF_i = \overline{a}_i/\overline{y}_i$. In a fair machine, these $AAF_i$ points would be closer together, as the $\overline{y}_i$ span is small. This idea is further elaborated in section 4.

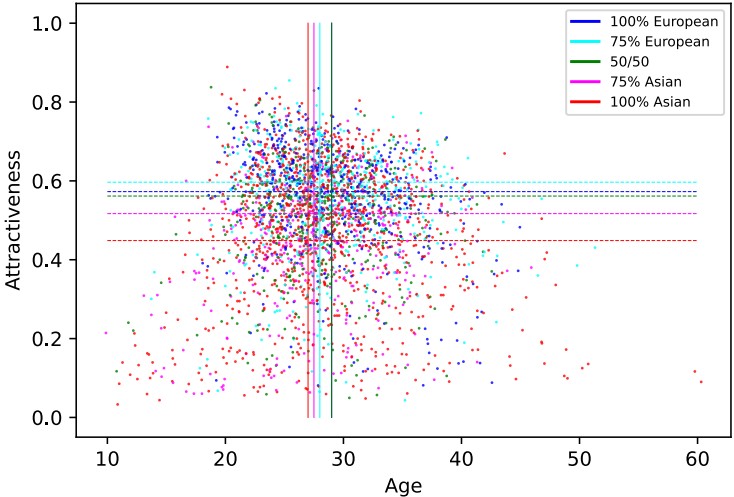

Figure 6: Biased correlation between attractiveness, age and ethnicity by German annotators. In an ethical, fair network the attractiveness for equal age groups would be the same. This would be represented in the figure by the same height of the lines for equal age groups.

## 4.2 TRAINING AND DATA PRE-PROCESSING

In our first approach of training the network we have applied pre-processing and resampling to the input data, which is explained in the following paragraphs.

This paper proposes a way to create a fair network with this biased data. Therefore, the bias must be identified in the ground truth labels of the dataset and divided into two subsets. The first subset (German annotations) confirms and increases the existing bias whereas the second subset (Chinese annotations) consists of the contrary prejudices. Afterwards, a GAN then generates synthetic images, which are a gradation of the mixture of the first and second subset. The least biased result according to our understanding is the best balance of the generated images. This knowledge can then be applied back to the original data set. This implies the height difference of all the bars should be minimised.

In our training process, we have a clear bias in the annotations, as shown in fig. 2 and measured in the analysis of the data. If we train our network based only on this labels, it follows the data and replicates the bias from the annotations, as shown in the comparison of the predictions with the annotations in fig. 2. Chinese annotators rate Asian faces higher, based on this data our prediction is biased towards higher aesthetic scores of Asian Faces. This is the same if we train the machine only on European Faces, annotated by Germans. In the next training, we added the annotations from the Chinese and German annotators and trained the network on an equal distribution of those annotations (Ratio: 1.0). The result is shown in fig. 7 on the left side of the diagram. The average aesthetic rating of European and Asian faces is still biased, however not as strong as in the previous experiment. The eleven bars on the right side of fig. 7 show the average aesthetic score based on the ethnicity. The bias is shown in more detail, ranging from 100% to 70% European who have the highest aesthetic score, to the lowest aesthetic score, the more Asian looking the portrait is. As a result, the network is still biased, a network trained on this data reflects the bias in the FBP.

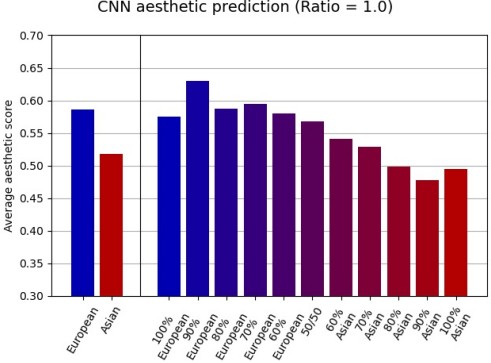
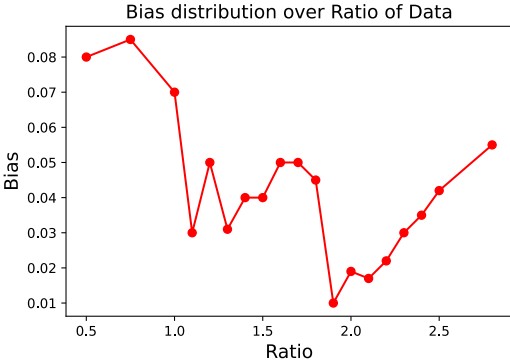

Figure 7: FBP with same amount of Asian and German labels. Ratio = 1.0 stands for the same weight $\omega$ for German and Asian annotations.

Figure 8: Correlation of the bias over the ratio of German and Chinese annotations. The least bias here is at the ratio of 1.9

In this experiment, balancing the training data means to find the minimum by concatenating the German annotated subset $g$ with the weighted $\omega$ Chinese annotated subset $c$. The goal in this approach is to level the average aesthetic scores $\overline{g}$ and $\overline{c}$ for the generated predictions $g_i$ and $c_i$. The network bias $B$ is then defined by

$$B = \frac{1}{2n+1} \sum_{i=0}^{n} |\overline{g} - g_i| + \omega \, |\overline{c} - c_i|. \tag{1}$$

Starting from a ratio of 1:1, in which German and Chinese annotations are distributed equally, we gradually increase the weight of the Chinese annotations. Technically, the balancing of distribution of the training data is done with a factor based approach. First, the ratio between Chinese and European annotations are calculated. Secondly, the factor for the balanced distribution is determined in a stochastic approach. In our experiment we varied the ratio from 2:1 to 1:3.2 for German annotations to Chinese annotations. Each training step and the corresponding bias over the ratio is shown in fig. 8. Determining the minimum in fig. 8 is equal to finding the least biased network. It is visible that a ratio of 1:1.9 produces the least biased network for this experiment and its results are shown in fig. 9. This means the Chinese annotations are weighted nearly double the amount than the European annotations.

Limitations of this approach are that information about the structure of the underlying latent features are unknown and balancing the network requires a lot of time and work. Therefore, we additionally propose another approach, described in the following section.

### 4.3 DEBIASING NEURAL NETWORK

#### 4.3.1 TRAINING NETWORK FEATURES

Regular convolutional neural networks (CNN) are generally used for face recognition tasks and we also used CNNs for FBP. They can be used to classify identities, and in our case to classify aesthetic scores (Serengil & Ozpinar, 2020), commonly called Facial Beauty Prediction (FBP). FBP using Machine Learning and Artificial Intelligence has been researched and improved many times in the past by various researches (Eisenthal et al., 2006; Gerlach et al., 2020; Kagian et al., 2008; Xie et al., 2015b; Liang et al., 2018; Liu et al., 2016; Xu et al., 2019).

Common for all those studies is that data is often generated by subgroups, with their own characteristics and behaviours (Mehrabi et al., 2019), especially in the highly subjective field of aesthetic rating. Therefore, the possibility exists, that all datasets are affected by bias, which the networks trained on them transfer into the FBP. Solving this problem, training an unbiased network with biased data, is a recently much discussed subject in the area of Machine Learning (Amini et al., 2019; Bellamy et al., 2018).

To achieve the first results on unbiased aesthetic estimation, we used the existing VGG-Face framework in Keras with TensorFlow and adjusted it. The network consists of 11 blocks, each containing a

linear operator and followed by one or more non-linearities such as ReLU and max pooling (Parkhi et al., 2015). We apply transfer learning here and use the pretrained model for Face Recognition (Parkhi et al., 2015). Building up on the face recognition, attractiveness estimation is similar to age estimation (Gyawali et al., 2020) performed by observing the facial features from portraits. Comparable to age estimation, the network then assigns the Portrait a beauty score.

The convolutional layers in the network are followed by a rectification layer (ReLu) as in (Krizhevsky et al., 2017). We used the Adam optimizer (Kingma & Ba, 2017). The input to our network is a face image of the size $256 \times 256 \times 3$, and it uses Zero-Padding around the edges, to ensure that the image information on the edge is not lost. Our input data is split into 60% train and 40% test data. The convolutional layers parameters of VGG-Face are not changed and kept frozen during the training. We use a dropout of 50%, and as it is a regression problem our final layer must be the size of 1. To classify the aesthetic score, a softmax activation function is used in the final layer. As a loss metric, we use the mean squared error and to compare our networks we also calculate the Pearson correlation and the root mean squared error.

### 4.3.2 BALANCED TRAINING

The process and the effect of the ratio on the average aesthetic rating is shown in fig. 8. By modifying the ratio of the annotations a minimum is determined that illustrates the lowest difference between the average aesthetic prediction of Asian and European faces. This represents a specific loss function for our network that maps bias onto measurable values. To remove bias from our network, we calculate the difference between European and Asian aesthetic predictions and find the global minimum.

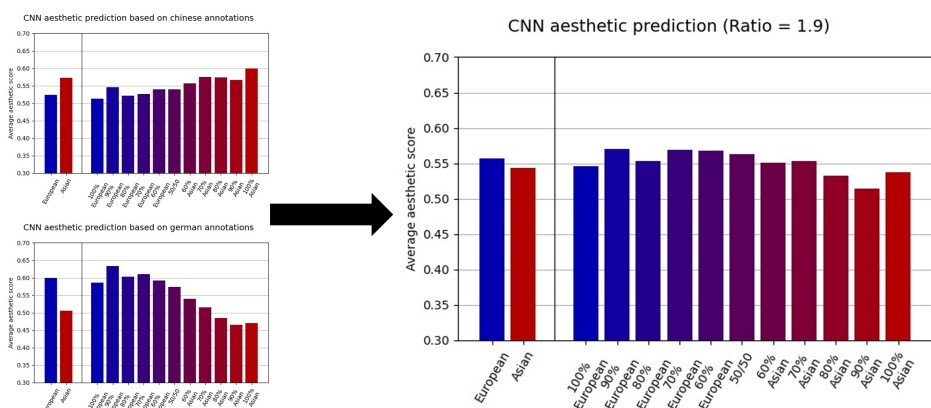

Figure 9: CNN aesthetic prediction with equalised distribution of training data. The charts on the left side show the prediction of the network if it is only trained on Chinese or German annotations. On the right side, the prediction of the network, which was trained on the biased data is shown. All bars have more or less the same height and only differ minimally. This means, that we could eliminate most of the bias in the training data, by balancing and we can assume that this trained network is fair.

The minimum of the average aesthetic score between Asian and European faces is located at a ratio of 1:1.9 where the average aesthetic score differs by about 5%. We create a model with a fair performance over all classes of different ethnicities as shown in fig. 9. This proves, that by resampling and balancing the training data a less biased AI can be created. We are retaining the precision in FBP, as shown in table 1. This process creates a less-biased AI in FBP tasks.

Our results are displayed in fig. 9 where all bar charts have a similar height and the FBP score is considerably less biased. Not all bars have the exact same height, this is due to some background noise. Real world data usually contains noise which affects tasks such as classification in machine learning (Gupta & Gupta, 2019). This noise also affects our aesthetic prediction, however with those minor differences, we can consider our network as unbiased and therefore fair. As we use a factor based approach to multiply the annotation data, this noise is present over all ratios. Only the difference of the averages increases or decreases within the variations of the ratio.

## 5 CONCLUSION

Our two main contributions are AestheticNet and a new approach to bias-free machine learning tools. In this work, we have proposed to augment the SCUT-FBP dataset by synthesised GAN images and show that AestheticNet predicts facial attractiveness with higher correlation then competitive approaches. Then we utilise a novel learning strategy to minimise bias in networks. Unbiased networks are an important step towards a future, where more decisions are made by AI and therefore more lives are influenced by artificial intelligence - unbiased decision making is the foundation of ethical and moral values.

Bias-free decision making is a challenging problem in machine learning tasks, yet it yields the great potential to be one of the most significant strengths of an AI. We have shown a method to eliminate bias in facial attractiveness prediction and this method can be transferred to multiple similar networks.

Training an unbiased model on biased data is an important goal from a Machine Learning Perspective, as perfect, unbalanced data might be raw. Especially in the field of Aesthetic Judgement, it is important that the machine is able to realise the bias. By learning, how to act against this bias, we can scale this approach in the future on larger datasets in other areas. The algorithm is introduced by applying it to aesthetic judgement, but not limited to it. Further development and deployment of fair and unbiased AI systems is crucial for AI to be a social benefit for all and reduce algorithmic discrimination.

Implicit bias has always been a hot-spot in the field of psychology in the 21$^{st}$ century. With the intersection of disciplines, a series of moral and ethical issues arising from it have also attracted the attention of the philosophical field. Implicit bias is widespread and is in a silent way. It affects all aspects of our lives, even in the field of artificial intelligence, which is equally popular in the 21$^{st}$ century. In this article, we have verified the universality of implicit existence and artificial intelligence will copy human prejudice by proving the establishment of two hypotheses. On this basis, we have analysed the reasons for the existence of implicit bias from neuroscience and psychology. Next, we used experimental methods to systematically demonstrate how human implicit bias affects the decision-making of artificial intelligence and found a way to eliminate the implicit bias of artificial intelligence. Secondly, we improved the fairness of the algorithm from machine learning. From a perspective, the training of unbiased models on biased data is an important goal, and unbiased networks are an important step towards the future. In the future, artificial intelligence will make more decisions, so more lives will be affected by artificial intelligence. Unbiased artificial intelligence decision-making is the moral foundation.

One problem that has to be tackled in the future is the issue that our attractiveness was elo rated and is hence not equally distributed per level of attractiveness. Very attractive and very unattractive faces are much less common than average faces. In order to make up for that we used stargan_v2 (Choi et al., 2020) to enhance our data set by computer generated faced that are meant to be more attractive than real faces. The generated images were rated in another survey and in fact came out to be more attractive than real ones. As scientific AI-researchers, we require our work to maintain sufficient awe of nature and morality. This is our current and future work. As Kant said, "Two things fill the mind with ever new and increasing admiration and awe, the more often and steadily we reflect upon them: the starry heavens above me and the moral law within me."

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

# A  SUPPLEMENTARY MATERIAL

**Toolbox**   The proposed toolbox introduces a huge collection of machine learning and face analysis applications. Specifically, to evaluate the annotation data for discriminatory bias, we designed the application that contains dedicated modules to detect correlations inside the network. The toolbox itself consists of the following modules: (a) preparation of the annotated data, (b) statistics generator, (c) dataset administration, (d) pretrained convolutional neural network (CNN) for hair colour, hair style, skin complexion, (e) unbiased, non-discriminatory aesthetic scores, (f) facial landmark detector, (g) 3D morphable model fitter.

**Pearson Correlation Coefficient**   The Pearson correlation coefficient is a value between -1 and 1, which is used to measure the correlation (linear correlation) between two variables X and Y; when one variable increases, the other variable also increases, indicating that there is a positive correlation between them, and the correlation coefficient is greater then 0; if one variable increases, the other variable decreases, indicating that there is a negative correlation between them, and the correlation coefficient is less than 0; if the correlation coefficient is equal to 0, there is no linearity correlation relationship, if the correlation coefficient is equal to 1, it means that they are linearly equal, that is, the scoring results after machine learning are completely equivalent to the artificial results of the experimental participants in China and Germany.

The Pearson Correlation Coefficient (PC) is a statistic that measures linear correlation between the annotation and the FBP of AestheticNet. The value of the PC is in the range of 1 to -1. 1 or -1 means there is a high linear correlation. 0 means there is no correlation.

$$r_{xy} = \frac{\sum_{i=1}^{n}(x_i - \bar{x})(y_i - \bar{y})}{\sqrt{\sum_{i=1}^{n}(x_i - \bar{x})^2}\sqrt{\sum_{i=1}^{n}(y_i - \bar{y})^2}} \tag{2}$$

**nMAE**   To compare different approaches we needed to normalise the errors. With this normalisation we are able to compare between datasets or models with different score ranges (e.g. beauty score from 1-5 or 1-7) with ours, as we used a score range from 1-10. The error is expressed as a percentage, lower values indicate less residual variance. In our case a lower nMAE or nRMSE indicates a higher prediction accuracy on the dataset. The average of mean error is normalised over the total score range (s).

$$nMAE = \frac{\sum|f_t - a_t|}{s_{max} - s_{min}} \tag{3}$$

**nRMSE**   The RMSE is defined as the square root of the mean square error. It is a standard way to measure the error of a model in predicting quantitative data.

$$RMSE = \sqrt{\frac{\sum_{i=1}^{n}(\hat{y}_i - y_i)^2}{n}} \tag{4}$$

Though there is no consistent means of normalisation in the literature, our choice is to mean the range of the measured data, which is the most common choice in normalisation. We divide the RMSE by the score range (s).

$$nRMSE = \frac{RMSE}{s_{max} - s_{min}} \tag{5}$$

The ranges of values in the "annotations" columns in table 2 are defined as

(i)   age: *years (int)*

(ii)   gender: *{0,1}*

(iii)   ethnic: *[0,1]*

(iv)   height: *meters (float)*

(v)   weight: *kilograms (float)*

(vi)   sports: {*Alpine Skiing, Biathlon Bobsleigh, Cross Country, Curling, Figure Skating, Freestyle Skiing, Ice Hockey, Luge, Short Track, Skeleton, Ski Jumping, Snowboard, Speed Skating*}

(vii) glasses: {*0,1*}

(viii) attractiveness: *[0,1]*

(ix) complexion: {*fair, medium, olive, deep*}

(x) hair colour: {*blond, light brown, brown, dark brown, black*}

(xi) hair style: {*pony, short, long, occlusion eye, occlusion cheek*}

To measure a bias, we evaluate our network on our database of synthetically created images (GAN), called the Eurasian dataset. In this way we are able to classify the ethnicity exactly and evaluate the difference of aesthetic score, being our bias. The idea of this experiment is that in a fair network, ethnicity does not affect the aesthetic rating. In a fair network all lines should be at the same height for the same age class, as mentioned in fig. 6.

## B  INTRODUCTION TO ETHICS OF AI

This section is a addition to the introduction of this work. While it does not provide information about the technical issue, it provides an introduction the ethics that drive the development of AestheticNet.

### B.1  MORAL ENHANCEMENT THROUGH AI

Faced with an increasingly diversified global environment, racial, religious, and cultural conflicts continue, and human morality is also facing unprecedented challenges. Many scholars have put forward the idea of human moral enhancement through technology, such as biotechnology, these interventions include the use of various substances, such as oxytocin and serotonin, as well as of various techniques, including transcranial magnetic stimulation and the provision of neurofeedback (Kelemen et al., 2015; Lara & Deckers, 2020).

The aim of these interventions would be to promote trust in others and to foster the desire to collaborate, but the harm that biotechnology does to the human body has always been a matter of concern. Many of us are biologically predisposed to have limited cognition and levels of altruism (Baron-Cohen, 2012; Wallace et al., 2007).

Everyone has an implicit bias, although this is not entirely caused by biological reasons. The existence of prejudice will lead to the problem of bias in machine learning, which has been proved in previous experiments. Compared with biotechnology, artificial intelligence is more trustworthy in helping humans with moral enhancement while it is improving the quality of human life. Especially existing implicit bias interventions tend to produce limited effects. For example, an air quality detection system installed on the road can remind us whether we should limit the number of private car trips and reduce gas emissions (Lai & J., 2016). An intelligent virus tracking APP reminds you that you might have come into contact with high-risk groups and that it is now necessary to consider home isolation. Some researchers are very pessimistic about the moral nature of humans while Dietrich is very optimistic about the possibilities of AI and believes that robots would have achieved that "Copernican turn" inaccessible to most humans by their biological conditioning (Dietrich, 2011). AI could monitor physical and environmental factors that affect moral decision-making, could identify and make agents aware of their biases, and could advise agents on the right course of action, based on the agent's moral values (Kelemen et al., 2015).

### B.2  SOCIAL PROBLEMS CAUSED BY AI

Machine learning dominated by human training inevitably enables artificial intelligence to replicate human bias, including human implicit bias. Implicit bias is a bias that may be unconscious or uncontrollable. It exists in almost everyone, and the resulting social problems are endless. These subconscious thoughts are learned through our experience and are so deeply rooted that we may ignore them. Temporary unconscious bias may be the source of discriminatory and discriminatory practices. For example, even though Americans consider themselves unbiased when measuring unconscious stereotypes 90% of whites (Moule, 2009) and 50% of blacks associate negative characteristics with black images. From the beginning of childhood, white children and children of colour have liked white dolls (News, 2009).

In 2010, child psychologist and University of Chicago professor Margaret Beale Spencer (leading researcher in the field of child development) was hired by CNN as a consultant. Her team tested 133 children from schools with specific economic and demographic requirements. Tests have shown that white children have a high reaction rate to "white prejudice". They identify the colour of their skin as positive, while darker skin is identified as negative. Dr. Spencer said that even black children are prejudiced against whites, but far fewer than white children (Jill Billante et al., 2010).

According to a Reuters report in October 2018, Amazon's AI recruiting tool discriminates against women (Dastin, 2018). The research and development team at Amazon has been creating applications since 2014 to check the performance of applicants using the trained AI recruitment engine. Despite many years of experience and an exceptionally long maturation period, such obvious disadvantages are difficult or even impossible to recognise. Judging from the resumes sent to Amazon by the AI recruitment engine over the past ten years, most of them are men. The system automatically lowered the ranks of the two women's universities and associated the keyword "women" with "captain of the women's chess club". As one of the nine major AI companies that go hand-in-hand with Google and Facebook, the AI discrimination scandal that Amazon has fallen into has aroused considerable repercussions in the field of artificial intelligence and has aroused heated discussions from all walks of life.

On May 25, 2020, in Minnesota, USA, a black man died after being kneeled on by police for 7 minutes (Hill et al., 2020). In an interview with CBS, Minneapolis Mayor Jacob Fry said, "I don't know if there is explicit or implicit racism, but racism is definitely involved".

As we needed a larger dataset for image annotation, we use recently published generative adversarial networks (GAN), such as StarGAN-v2 (Choi et al., 2020). For our annotation process we create portraits of females and manually selected outcomes with no or only very few artefacts. In the annotation process, the annotators label a mix of GAN images and real images from the SCUT FBP (Liang et al., 2018) dataset. After we evaluate the results of the annotation, we realise that the highest annotations are dominated by images generated with StarGAN. With our dataset, we experience no uncanny valley problems. We can assume that during clicking the border between real person and synthetically generated person blurs. This is also demonstrated on the website *thispersondoesnotexist.com* with the accompanying publication by Karras et al. (Karras et al., 2019). In addition, generative networks producing synthetically generated portraits and animations, called deepfakes, are currently on the rise. Simultaneously, due to current circumstances, meetings using video-calls are currently on the rise. It could be argued that soon, within a video call, it will not possible to differentiate between a real human being or a deepfake of a person by vision alone. This creates many more questions on social problems which might be created by AI.

## C  WHAT IS UNCONSCIOUS BIAS?

Obviously, unconscious bias is a kind of prejudice that can be expressed as positive preference or negative discrimination. Favouritism is morally acceptable in many cases. A preference for both one's own inner group, such as family and friends, and for the support of a weak social group usually does not trigger the exclusion of external groups. The preference in this case did not lead to obvious social problems. However, if the prejudice encompasses a large, specific group, such as ethnicity, gender, or age, then moral and ethical problems are more likely to arise.

Another situation is that when the result of preference leads to unfair treatment of a certain individual or group, discrimination also occurs, and its counterpart is explicit bias. Inner group preference is a common cause of unconscious bias, but as explained earlier, it is also likely to lead to discrimination.

Unconscious bias is often used interchangeably with implicit bias in the fields of philosophy and psychology. Because unconscious bias is literally easier to understand and accept by the general public, it is used more often in a wider range of everyday language. Implicit bias was first defined by psychologists Mahzarin Banaji and Anthony Greenwald in 1995, where they argued that social behaviour is largely influenced by unconscious associations and judgements, corresponding to it is explicit bias.

However, most psychologists have abandoned Freud's psychoanalytic theory of unconscious mental processes. Unconscious bias usually manifests as a stereotype of things. It manifests in many forms and often occurs in our daily lives. The most common manifestations are prejudices and stereotypes

about affinity, gender, or the appearance of people. For example, the unconscious bias of the title professor is automatically assigned to the portrayal of an older man, although there are also many young and female professors. In addition, many people have the stereotype that women are worse at math problems and better at verbal problems than men (Johns et al., 2005).

Affinity bias refers to when you unconsciously prefer people who share qualities with you or someone you like. This bias was also verified in our first part of the experiment. Networks trained by Europeans think European faces are more beautiful than Asian faces, in contrast, networks trained by Asians think Asian faces are more beautiful than European faces. A study at Yale shows the "male" candidate was judged to be more talented and experienced; he was selected for the job more often and at a higher salary (Moss-Racusin et al., 2012).

Obviously, this is a gender bias. And when you unconsciously notice people's appearances and associate it with their personality, you might have beauty bias. Untidy appearance does not mean that this is a person who lacks the ability to manage themselves. A person in shabby clothes may be economical but he is not necessarily poor. Unconscious biases like these happen to everyone every day. And as proved later in our experiment, the bias in the training data is likely to be given to the artificial intelligence, which replicates the bias.

## C.1 CAUSES OF UNCONSCIOUS BIAS

**1. Susceptibility to bias.** We are used to the brain's fast, emotional, unconscious thinking mode. Kahneman states in (Kahneman, 2011) that there are two systems in the brain to organise our daily life: System 1 and System 2. Fast, emotional, and unconscious activities like driving, talking, or cleaning use System 1 since it requires little or even no effort, but it is often prone to errors. System 2 is slow, logical, effortful, conscious thought, where reason dominates. System 1 is a kind of mental shortcut, and we take this shortcut. Rules of thumb, educated guesses, and using "common sense" are all forms of mental shortcuts. Implicit bias is a result of taking one of these cognitive shortcuts inaccurately (Rynders, 2019). As a result, we incorrectly rely on these unconscious stereotypes to provide guidance in a very complex world. Especially when we are under high levels of stress we are more likely to rely on these biases than to examine all the relevant surrounding information (Wigboldus et al., 2004).

**2. We seek patterns.** One key reason we develop such biases is that our brains have a natural tendency to look for patterns and associations in order to make sense of a very complicated world. Research shows that even before kindergarten, children already use their group membership (e.g., racial group, gender group, age group, etc.) to guide inferences about the psychological and behavioural traits. At such a young age, they have already begun to seek out patterns and recognise what distinguishes them from other groups. Not only do children recognise what sets them apart from other groups, they believe "what is similar to me is good, and what is different from me is bad" (Cameron et al., 2001). Children aren't just noticing how similar or dissimilar they are to others, but also that dissimilar people are actively disliked (Aboud, 1989). Recognising what sets you apart from others and then forming negative opinions about those outgroups (a social group with which an individual does not identify) contributes to the development of implicit biases.

**3. Social and cultural influences.** Influences from media, culture, education, and your individual upbringing can also contribute to the rise of implicit associations that people form about the members of social outgroups. Media has become increasingly accessible, and while that has many benefits, it can also lead to implicit biases.

The way TV portrays individuals, or the language journal articles use, can ingrain specific biases in our mind. They can lead us to associate Black people as criminals or females as nurses or teachers. How children are raised can also play an important role. One research study found that parental racial attitudes can influence children's implicit prejudice (Sinclair et al., 2005). Parents are not the only figures who can influence such attitudes. Siblings, the school setting, and the culture in which you grow up can also play a role in shaping your explicit beliefs and implicit biases.

Social education also has a powerful effect because it includes not only traditional school education but also family education and self-study. Learning is the process of acquiring new understanding, knowledge, behaviour, skills, values, attitudes, and preferences (Gross, 2010).

From a blank sheet of paper at birth to receiving education and learning, gaining knowledge and gradually forming their values and understanding of surrounding things, there is no doubt that education and learning play a vital role in a person's will and thinking. A common belief is that education is an important determinant to racial prejudice, and there is preliminary evidence that the effect of this education varies from country to country (Hello et al., 2002).

Research scholars have found that the Polish education system plays a decisive role in the nationalism and prejudice of students (Żuk, 2018). In the medical field, because the attitude of medical staff to obese individuals has contributed to discrimination and led to poor health, the medical education environment may have explicit and implicit biases against obesity. Researchers who have adopted innovative educational interventions (read about obesity drama) found that it has a significant effect on reducing implicit prejudice against obese people (Matharu et al., 2014).

## C.2 Influence on Artificial Intelligence

Numerous studies show that human-trained machines repeat human bias (Zuiderveen Borgesius, 2018; Stephens-Davidowitz, 2014; Munger, 2017; Horvitz, 2017). The assumption is to create unbiased artificial intelligence by inserting labelled data from people who are free of prejudice. Unfortunately, everyone is biased because a large part of human bias is unconscious and hard to detect. Unconscious prejudices affect 90 to 95 percent of people. Psychologists demonstrated this at a press conference at the University of Washington and presented a new tool that measures the unconscious roots of prejudice (Schwarz, 1998).

All science suffers from human bias. Even if we train giant robots to collect, store, and manipulate data for us, the ultimate observers, in the final analysis interpreters, and mediators of that data, are humans (Mukherjee, 2016).

Unconscious bias comes from the education background, the culture, attitudes, and stereotypes we pick up from the world we live in, and research over time and from different countries shows that it tends to line up with general social hierarchies. In words, unconscious bias is part of human nature and affects everyone, whether they realise it or not. Therefore, we can only try to avoid it and work with biased data; there is no way to completely eliminate it.

What is the significance of our research for the society? Firstly, from the field of image vision that we are engaged in, AI bias also exists in the field of image processing. Artificial intelligence applications tag pictures of White American brides as "brides", "dresses", and "weddings" while pictures of North Indian brides are tagged as "performing arts" and "costumes" (Shankar et al., 2017). Angwin's and Larson's (Larson & Angwin, 2016) analysis of ethical bias has prompted research showing that the disparity can be addressed if the algorithms focus on the fairness of outcomes. That which applies automatic labels to pictures in digital photo albums, was classifying images of black people as gorillas (BBC, 2015). Since the data set does not contain enough ethnic minorities, the artificial intelligence judges which designed by beauty. AI does not like black-skinned women (beauty.ai, 2016). Just imagine this would be a job position for a cover model position: there is no doubt that candidates with black skin will be rejected.

Although both Hume (David, 1898) and Kant (Kant & Guyer, 2000) believe that aesthetic judgement is only a subjective feeling which regarding the pleasure that we take from a beautiful object, and aesthetics itself is neither right nor wrong nor moral, the behaviour caused by aesthetic judgements is related to morality (Cui et al., 2019; Haidt, 2001). Many studies show that unfair recruitment cases are encountered due to aesthetic judgements (Maddox & Perry, 2018; Mason, 2017; Beattie & Johnson, 2012). More attractive people have higher incomes than less attractive individuals (Anýžová & Matějů, 2018; French, 2002; Cawley, 2004). More and more companies use AI enhanced recruitment systems, such as Facebook, LinkedIn and Unilever. The fairness of the image processing system is very important to the entire AI recruitment system. Our research provides a practical example of how to build a fair and unbiased AI.

Furthermore, through this research, we hope the unconscious bias will get more social attention by studying how unconscious bias affects decision-making in artificial intelligence. What is gratifying is that many companies are beginning to pay attention to the negatives that unconscious bias may entail and are making continuous efforts. Examples are Google (Google LLC, 2016) and Facebook (Facebook Inc., 2014). They are strengthening the training of employees in this area. Facebook

Table 3: Comparison of prediction accuracy on SCUT-FBP5500

|  | PC | nMAE (%) | nRMSE (%) |
|---|---|---|---|
| AlexNet (Liang et al., 2018) | 0.8298 | 7.345 | 9.548 |
| AlexNet (Zhai et al., 2019) | 0.8634 | | |
| ResNet-18 (Liang et al., 2018) | 0.8513 | 7.045 | 9.258 |
| ResNeXt-50 (Liang et al., 2018) | 0.8777 | 6.295 | 8.313 |
| HMTNet(Xu et al., 2019) | 0.8783 | 6.2525 | |
| 8.158 AaNet (Lin et al., 2019) | 0.9055 | 5.590 | 7.385 |
| P-AaNet (Lin et al., 2019) | 0.8965 | 5.713 | 7.588 |
| 2M BeautyNet (Gan et al., 2020) | 0.8996 | | |
| EfficientNetB3 based AestheticNet (ours) | 0.9011 | 5.841 | 7.663 |
| VGG-Face based AestheticNet (ours) | 0.9363 | 4.400 | 6.261 |
| **AestheticNet (ours)** | **0.9601** | **3.896** | **5.580** |

designed a webpage to make unconscious bias training videos widely available and Google has put about 60,000 employees through a 90-minute unconscious bias training program. This will help reduce human bias during human-computer interaction, but the economic cost of the training is also huge. If there is an artificial intelligence similar to our research that can better circumvent human unconscious biases it would be a viable alternative.

# D  TRAINING

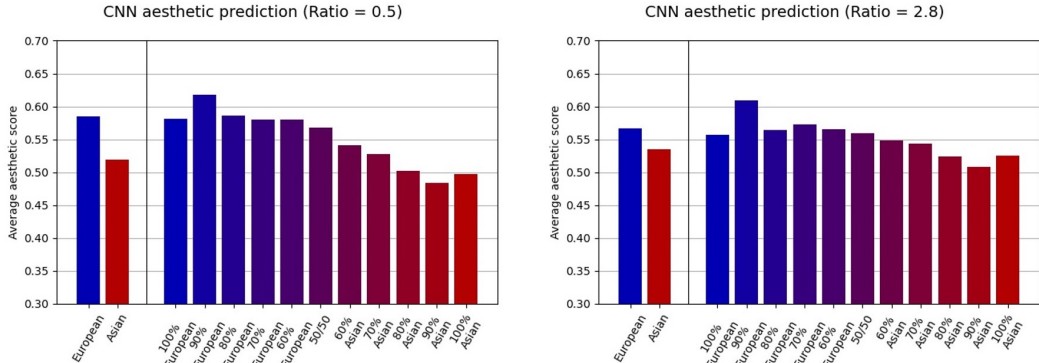

Figure 10: Effect of different ratios on the output of the network.

## D.1  EVALUATION OF UNBIASED NETWORK

We compare our method with other state-of-the-art approaches on the SCUT-FBP500 datasets. As shown in table 3 our AestheticNet therefore significantly surpasses previous approaches, which is mainly due to the augmentation with synthetic images and the optimisation of the previous approaches. In our best experiment, we achieve a Pearson correlation of 0.9601, a normalised mean average error of 3.896% and a normalised root mean squared error of 5.580%. The results are normalised because there are different datasets with different score ranges.

Having a state-of-the-art aesthetic prediction network, we then train a third CNN on the features from the Asian and German labelled networks to generate a non-biased network. Therefore, the synthesised Eurasian dataset is used with a categorical-cross-entropy-loss-function to converge the subgroup's intersection points of fig. 6.

### D.1.1  REMOVING BIAS USING CLUSTERED LABELS

A more sophisticated approach in getting rid of the bias in training data is our second approach. Within this we are developing a new method to reduce the bias in the training data. This method

consists of a deep learning network that is trained on the original learning task within the data set, and then minimises the bias inside the learned latent distributions using a specially adapted loss function.

Each data record contains a list of labels $a = a_1, ..., a_n$, which are to be debiased, and a further list of labels $b = b_1, ..., b_n$. In this example we remove the bias from the ethnical label $a_1$ and preserve the age, profession, hair colour and skin complexion labels. The network evaluates all attributes of the data set during the training and groups all objects according to the attributes $b$ in clusters.

Within each subgroup the difference between the ethnical mean value $\overline{a_1}$ represents the bias. A nonlinear operation, similar to the gamma correction in image systems, is then applied to the ethnic label to preserve the range of the values and bring the differences closer together. These differences for all clusters are the measure of the loss function, which is implemented as categorical cross entropy loss and should be minimised during training. With this we present a universally adaptable method to make any network fairer according to given labels.

