# OpenReview forum: "AestheticNet: Reducing bias in facial data sets under ethical considerations"
_ICLR.cc/2022/Conference — ICLR 2022 Submitted_

### Official Review · Reviewer_8hSi · 2021-10-27

**Correctness:** 3
**Technical Novelty And Significance:** 1
**Empirical Novelty And Significance:** 1
**Recommendation:** 1
**Confidence:** 4

**Main Review:**

In this paper, the authors study the problem of bias in facial beauty prediction problem. To this end, they first show that there do exist bias in an existing dataset. Then, they show that deep networks trained with such a biased dataset do capture and reflect the bias. Finally, they propose two solutions for addressing such bias.

Strengths:
- Bias is an important problem.
- Extensive analysis is provided.

Weaknesses:
- The most critical issue with the paper is the novelty. Neither the results or the insights nor the proposed methods are new or adding anything on top of the existing findings or solutions.

- The paper should utilize existing bias and fairness metrics from the literature and prefer to use such metrics for discussing the results instead of ambiguous descriptions.

Minor comments:
- The second paragraph of the abstract feels somehow disconnected with the first one. In any case, the abstract should provide more information about the experiments & the obtained results regarding fairness.
- Figures 1 and 2 are way too small to comprehend.
- "Figure 1: Chinese and European annotations" => The legend uses "Caucasian" instead.
- "to prove whether hypothesis 1 is true" => "Prove" is a very strong word. I would suggest "support".
- "Figure 1 and fig. 2" => Please be consistent.
- "Our training results have a very high accuracy and outperform state-of-the-art results." => I think you should refer to Table 1 here.


**Summary Of The Paper:**

In this paper, the authors study the problem of bias in facial beauty prediction problem. To this end, they first show that there do exist bias in an existing dataset. Then, they show that deep networks trained with such a biased dataset do capture and reflect the bias. Finally, they propose two solutions for addressing such bias.

**Summary Of The Review:**

Novelty is unclear. The paper just confirms existing findings and does not introduce any novel method.

---

### Official Review · Reviewer_V5EV · 2021-11-01

**Correctness:** 2
**Technical Novelty And Significance:** 1
**Empirical Novelty And Significance:** Not applicable
**Recommendation:** 1
**Confidence:** 5

**Details Of Ethics Concerns:**

I believe there exists anonymity issue in this work.

In Section 1 (Motivation), the authors write :

"Our data analysis has already proven that people consider their own ethnicity to be more attractive than others (Gerlach et al., 2020), this is the major bias in our experiments and within our dataset."

Also, Figure 1 is copied from Figure 10 and 11 from (Gerlach et al 2020).

Reference (Gerlach et al 2020): Tobias Gerlach, Michael Danner, Le Peng, Aidas Kaminickas, Wu Fei, and Matthias Ratsch. ¨ Who Loves Virtue as much as He Loves Beauty?: Deep Learning based Estimator for Aesthetics of Portraits:. In Proceedings of the 15th International Joint Conference on Computer Vision, Imaging and Computer Graphics Theory and Applications, pp. 521–528, Valletta, Malta, 2020. SCITEPRESS - Science and Technology Publications. ISBN 978-989-758-402-2. doi: 10.5220/0009172905210528. URL http://www.scitepress.org/DigitalLibrary/ Link.aspx?doi=10.5220/0009172905210528.

**Main Review:**

There is severe anonymity issue in this work. Also there is overlap with an existing work (Gerlach et al 2020).

Please see 'ethics concern'  for more details.

**Summary Of The Paper:**

The paper proposes a method to build an unbiased CNN for facial beauty prediction.

**Summary Of The Review:**

I'm rejecting the paper because of breach of anonymity. Moreover, the work lacks novelty. Please see 'ethics concern'  for more details.

---

### Official Review · Reviewer_txuX · 2021-11-02

**Correctness:** 3
**Technical Novelty And Significance:** 3
**Empirical Novelty And Significance:** 3
**Recommendation:** 6
**Confidence:** 4

**Main Review:**

Strengths:
1. The proposed AestheticNet achieves state-of-the-art performance on SCUT-FBP5500 and is significantly better than the competition.
2. The proposed bias-free machine learning tools consist two parts: “balanced training” and a categorical cross entroy loss function. The experimental results show that these two reach the goal of unbiased predictions.
Weaknesses:
1. The proposed network has not been elaborated and lacks some ablation experiments.
2. The judgment for unbiased networks is based on the ability of the network to output similar average aesthetic scores for Asians, Europeans, and all mixed-racial portraits. However, since the faces of different subsets keep changing, how to ensure that the actual attractivenesses of these different subsets have similar unbiased scores?
3. There are typos (eg. “then” in the third line of the conclusion should be “than”), and the labels of the charts are too small (eg. the chart on the left side of Figure 9).

**Summary Of The Paper:**

This paper proposes an AestheticNet and a new approach to bias-free machine learning tools. The former shows a higher Pearson correlation coefficient and a lower mean absolute error than competitive approaches. The latter helps to train an unbiased network with biased data for facial beauty prediction.

**Summary Of The Review:**

The main contributions of this paper are AestheticNet and a new approach to bias-free machine learning tools. The results look sound but some details are not elaborated (See the main review for details). I think this paper is marginally above the acceptance threshold.

---

### Official Review · Reviewer_rA4x · 2021-11-02

**Correctness:** 1
**Technical Novelty And Significance:** 1
**Empirical Novelty And Significance:** 1
**Recommendation:** 1
**Confidence:** 5

**Details Of Ethics Concerns:**

Prediction of beauty is a potential harmful application. Only women are evaluated in this study. There are too many biases in "beauty" prediction for a system to solve as the paper claims.

**Main Review:**

- Motivation for this type of work is lacking. What can be gained from beauty prediction, even if it is unbiased?
- Hypothesis 1 is known, this is not a new hypothesis. The paper cites works (Gerlach et al, Akbari) that show that hypothesis is true. It is not clear what is learned from H1.
- Hypothesis 2 is also known. This is one of the main problems with bias in AI. The human biases we have are in our AI systems. It is not clear what is learned from H2.
- Section 3.1 - What is the Asia-Europe dataset that is evaluated? What does this dataset have to do with the benchmark dataset used?
- Proposed architecture is not novel. It is a modified version of VGG Face. It is also not clear what these modifications are.
- Training based on GANs is not new and has been done extensively in AI/ML.
- Only using Chinese and German faces is bias in itself. What insight can be gained from this? As beauty is dependent on ethnicity, this work can not generalize to anything else.
- What is the definition of attractive, unattractive, and average face? This definition is inherently biased by ethnicity and age. How can it be defined here?

**Summary Of The Paper:**

Paper proposed aestheicNet in order to solve the problem of bias in beauty prediction.

**Summary Of The Review:**

There is no supporting motivation for this work.  The paper makes claims of debiasing "beauty" predictions. These claims are not supported. Two ethnicities are used, which is inherently biased. The hypotheses in the paper offer no new insight as the paper cites works that have shown similar before. If it is known the hypothesis is true, there is no need to test it.

---

### Decision · Program_Chairs · 2022-01-20

**Decision:**

Reject

**Comment:**

The paper proposes a new neural network, the aestheticNet, for a bias-free facial beauty prediction.
All the reviewers agree that the work is not suitable for publication as it raised some serious ethic concerns:
* Prediction of beauty (aesthetic scores) is a potential harmful application. Well-intended as it may be, a research along these lines might be harmful.
* non-anonoymity issue: writing reveals/implies authors identity with reference to previous work
* Research integrity issues (e.g., plagiarism, dual submission), a figure is copied from previous work.

There is also a concern that the work is not novel and not interesting as such.
The authors did not respond to the concerns.

I suggest rejection.